# TransGAN: Two Pure Transformers Can Make One Strong GAN, and That Can Scale Up

**Yifan Jiang**[1], **Shiyu Chang**[2,3], **Zhangyang Wang**[1]
[1]University of Texas at Austin
[2]UC Santa Barbara    [3]MIT-IBM Watson AI Lab
{yifanjiang97,atlaswang}@utexas.edu,  chang87@ucsb.edu

## Abstract

The recent explosive interest on transformers has suggested their potential to become powerful "universal" models for computer vision tasks, such as classification, detection, and segmentation. While those attempts mainly study the discriminative models, we explore transformers on some more notoriously difficult vision tasks, e.g., generative adversarial networks (GANs). Our goal is to conduct the first pilot study in building a GAN *completely free of convolutions*, using only pure transformer-based architectures. Our vanilla GAN architecture, dubbed **TransGAN**, consists of a memory-friendly transformer-based generator that progressively increases feature resolution, and correspondingly a multi-scale discriminator to capture simultaneously semantic contexts and low-level textures. On top of them, we introduce the new module of grid self-attention for alleviating the memory bottleneck further, in order to scale up TransGAN to high-resolution generation. We also develop a unique training recipe including a series of techniques that can mitigate the training instability issues of TransGAN, such as data augmentation, modified normalization, and relative position encoding. Our best architecture achieves highly competitive performance compared to current state-of-the-art GANs using convolutional backbones. Specifically, TransGAN sets the **new state-of-the-art** inception score of 10.43 and FID of 18.28 on STL-10. It also reaches the inception score of 9.02 and FID of 9.26 on CIFAR-10, and 5.28 FID on CelebA $128 \times 128$, respectively: both on par with the current best results. When it comes to higher-resolution (e.g. $256 \times 256$) generation tasks, such as on CelebA-HQ and LSUN-Church, TransGAN continues to produce diverse visual examples with high fidelity and reasonable texture details. In addition, we dive deep into the transformer-based generation models to understand how their behaviors differ from convolutional ones, by visualizing training dynamics. The code is available at: https://github.com/VITA-Group/TransGAN.

## 1 Introduction

Generative adversarial networks (GANs) have gained considerable success on numerous tasks [1, 2, 3, 4, 5, 6, 7]. Unfortunately, GANs suffer from the notorious training instability, and numerous efforts have been devoted to stabilizing GAN training, introducing various regularization terms [8, 9, 10, 11], better losses [1, 12, 13, 14], and training recipes [15, 16]. Among them, one important route to improving GANs examines their *neural architectures*. [17, 8] reported a large-scale study of GANs and observed that when serving as (generator) backbones, popular neural architectures perform comparably well across the considered datasets. Their ablation study suggested that most of the variations applied in the ResNet family resulted in very marginal improvements. Nevertheless, neural architecture search (NAS) was later introduced to GANs and suggests enhanced backbone designs are also important for improving GANs, just like for other computer vision tasks. Those works are consistently able to discover stronger GAN architectures beyond the standard ResNet topology

35th Conference on Neural Information Processing Systems (NeurIPS 2021).

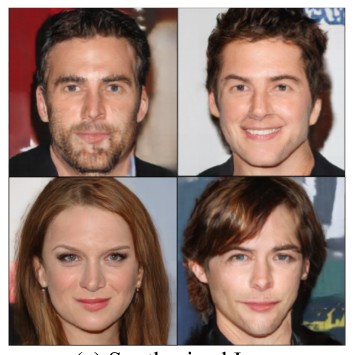 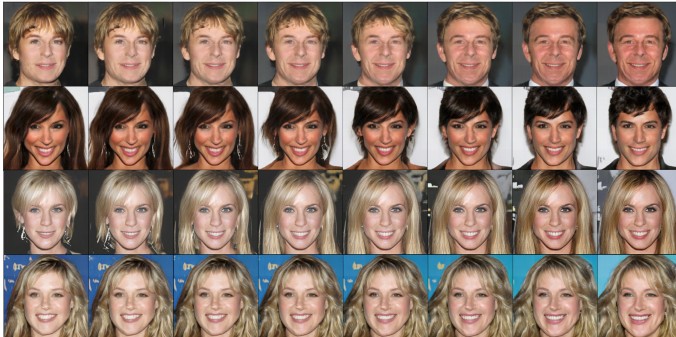

| (a) Synthesized Image | (b) Interpolation on Latent Space |

Figure 1: Representative visual examples synthesized by TransGAN, **without using convolutional layers**. (a) The synthesized visual examples on CelebA-HQ ($256 \times 256$) dataset. (b) The linear interpolation results between two latent vectors, on CelebA-HQ ($256 \times 256$) dataset.

[18, 19, 20]. Other efforts include customized modules such as self-attention [21], style-based generator [22], and autoregressive transformer-based part composition [23].

However, one last "commonsense" seems to have seldomly been challenged: using convolutional neural networks (CNNs) as GAN backbones. The original GAN [24, 25] used fully-connected networks and can only generate small images. DCGAN [26] was the first to scale up GANs using CNN architectures, which allowed for stable training for higher resolution and deeper generative models. Since then, in the computer vision domain, every successful GAN relies on CNN-based generators and discriminators. Convolutions, with the strong inductive bias for natural images, crucially contribute to the appealing visual results and rich diversity achieved by modern GANs.

***Can we build a strong GAN completely free of convolutions?*** This is a question not only arising from intellectual curiosity, but also of practical relevance. Fundamentally, a convolution operator has a local receptive field, and hence CNNs cannot process long-range dependencies unless passing through a sufficient number of layers. However, that is inefficient, and could cause the loss of feature resolution and fine details, in addition to the difficulty of optimization. Vanilla CNN-based models are therefore inherently not well suited for capturing an input image's "global" statistics, as demonstrated by the benefits from adopting self-attention [21] and non-local [27] operations in computer vision. Moreover, the spatial invariance possessed by convolution poses a bottleneck on its ability of adapting to spatially varying/heterogeneous visual patterns, which also motivates the success of relational network [28], dynamic filters [29, 30] and kernel prediction [31] methods.

## 1.1 Our Contributions

This paper aims to be the **first pilot study** to build a GAN completely free of convolutions, using only pure transformer-based architectures. We are inspired by the recent success of transformer architectures in computer vision [32, 33, 34]. Compared to parallel generative modeling works [21, 23, 35] that applied self-attention or transformer encoder in conjunction with CNN-based backbones, our goal is more ambitious and faces several daunting gaps ahead. First and foremost, although a pure transformer architecture applied directly to sequences of image patches can perform very well on image classification tasks [34], it is unclear whether the same way remains effective in generating images, which crucially demands the spatial coherency in structure, color, and texture, as well as the richness of fine details. The handful of existing transformers that output images have unanimously leveraged convolutional part encoders [23] or feature extractors [36, 37]. Moreover, even given well-designed CNN-based architectures, training GANs is notoriously unstable and prone to mode collapse [15]. Training vision transformers are also known to be tedious, heavy, and data-hungry [34]. Combining the two will undoubtedly amplify the challenges of training.

In view of those challenges, this paper presents a coherent set of efforts and innovations towards building the **pure** transformer-based GAN architectures, dubbed **TransGAN**. A naive option may directly stack multiple transformer blocks from raw pixel inputs, but that would scale poorly due to memory explosion. Instead, we start with a memory-friendly transformer-based generator by gradually increasing the feature map resolution in each stage. Correspondingly, we also improve the discriminator with a multi-scale structure that takes patches of varied size as inputs, which

balances between capturing global contexts and local details, in addition to enhancing memory efficiency more. Based on the above generator-discriminator design, we introduce a new module called *grid self-attention*, that alleviates the memory bottleneck further when scaling up TransGAN to high-resolution generation (e.g. $256 \times 256$).

To address the aforementioned instability issue brought by both GAN and Transformer, we also develop a unique training recipe in association with our innovative TransGAN architecture, that effectively stabilizes its optimization and generalization. That includes showings the necessity of data augmentation, modifying layer normalization, and replacing absolute token locations with relative position encoding. Our contributions are outlined below:

- **Novel Architecture Design:** We build the first GAN using purely transformers and no convolution. TransGAN has customized a memory-friendly generator and a multi-scale discriminator, and is further equipped with a new grid self-attention mechanism. Those architectural components are thoughtfully designed to balance memory efficiency, global feature statistics, and local fine details with spatial variances.
- **New Training Recipe:** We study a number of techniques to train TransGAN better, including leveraging data augmentation, modifying layer normalization, and adopting relative position encoding, for both generator and discriminator. Extensive ablation studies, discussions, and insights are presented.
- **Performance and Scalability:** TransGAN achieves highly competitive performance compared to current state-of-the-art GANs. Specifically, it sets the new state-of-the-art inception score of 10.43 and FID score of 18.28 on STL-10. It also reaches competitive 9.02 inception score and 9.26 FID on CIFAR-10, and 5.28 FID score on CelebA $128 \times 128$, respectively. Meanwhile, we also evaluate TransGAN on higher-resolution (e.g., $256 \times 256$) generation tasks, where TransGAN continues to yield diverse and impressive visual examples.

## 2    Related Works

**Generative Adversarial Networks.**    After its origin, GANs quickly embraced fully convolutional backbones [26], and inherited most successful designs from CNNs such as batch normalization, pooling, (Leaky) ReLU and more [38, 39, 40, 18]. GANs are widely adopted in image translation [3, 4, 41], image enhancement [7, 42, 43], and image editing [44, 45]. To alleviate its unstable training, a number of techniques have been studied, including the Wasserstein loss [46], the style-based generator [22], progressive training [16], lottery ticket [47], and spectral normalization [48].

**Transformers in Computer Vision.**    The original transformer was built for NLP [49], where the multi-head self-attention and feed-forward MLP layer are stacked to capture the long-term correlation between words. A recent work [34] implements highly competitive ImageNet classification using pure transformers, by treating an image as a sequence of $16 \times 16$ visual words. It has strong representation capability and is free of human-defined inductive bias. In comparison, CNNs exhibit a strong bias towards feature locality, as well as spatial invariance due to sharing filter weights across all locations. However, the success of original vision transformer relies on pretraining on large-scale external data. [50, 51] improve the data efficiency and address the difficulty of optimizing deeper models. Other works introduce the pyramid/hierarchical structure to transformer [52, 53, 54] or combine it with convolutional layers [55, 56]. Besides image classification task, transformer and its variants are also explored on image processing [37], point cloud [57], semantic segmentation [58], object detection [32, 59] and so on. A comprehensive review is referred to [60].

**Transformer Modules for Image Generation.**    There exist several related works combining the transformer modules into image generation models, by replacing certain components of CNNs. [61] firstly formulated image generation as autoregressive sequence generation, for which they adopted a transformer architecture. [62] propose sparse factorization of the attention matrix to reduce its complexity. While those two works did not tackle the GANs, one recent (concurrent) work [23] used a convolutional GAN to learn a codebook of context-rich visual parts, whose composition is subsequently modeled with an autoregressive transformer architecture.The authors demonstrated success in synthesizing high-resolution images. However, the overall CNN architecture remains in place (including CNN encoder/decoder for the generators, and a fully CNN-based discriminator), and the customized designs (e.g, codebook and quantization) also limit their model's versatility. Another concurrent work [35] employs a bipartite self-attention on StyleGAN and thus it can propagate latent variables to the evolving visual features, yet its main structure is still convolutional, including both the

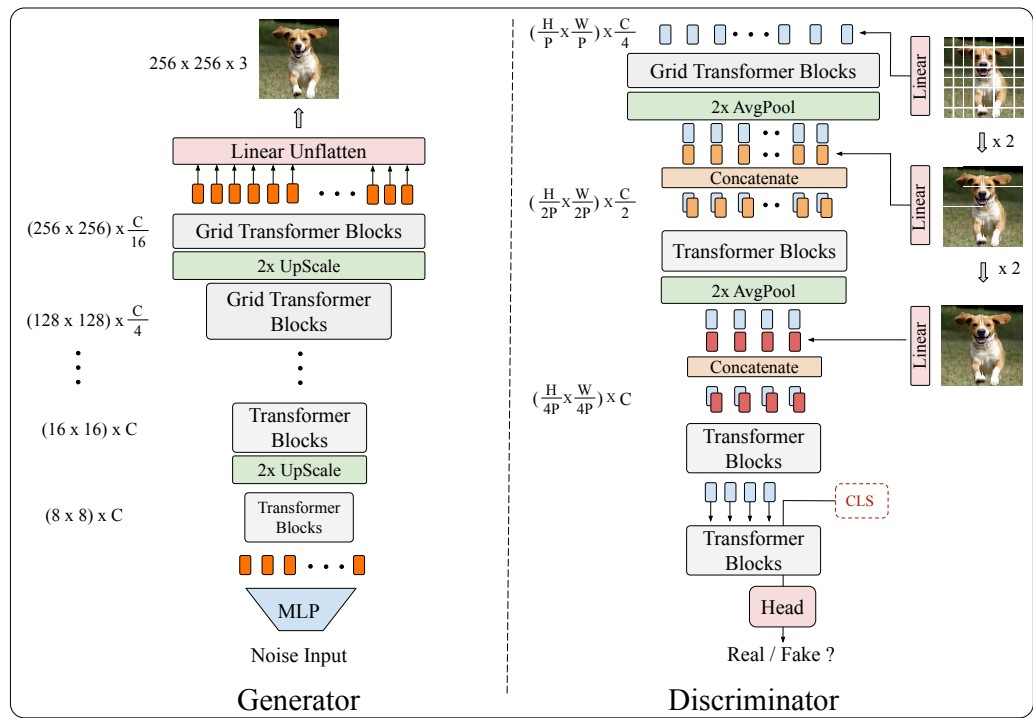

Figure 2: The pipeline of the pure transform-based generator and discriminator of TransGAN. We take $256 \times 256$ resolution image generation task as a typical example to illustrate the main procedure. Here patch size $p$ is set to 32 as an example for the convenience of illustration, while practically the patch size is normally set to be no more than $8 \times 8$, depending on the specific dataset. `Grid Transformer Blocks` refers to the transformer blocks with the proposed grid self-attention. Detailed architecture configurations are included in Appendix B.

generator and discriminator. To our best knowledge, no other existing work has tried to completely remove convolutions from their generative modeling frameworks.

## 3  Technical Approach: A Journey Towards GAN with Pure Transformers

In this section, we start by introducing the memory-friendly generator and multi-scale discriminator, equipped with a novel grid self-attention. We then introduce a series of training techniques to stabilize its training procedure, including data augmentation, the modified normalization, and injecting relative position encoding to self-attention.

To start with, we choose the transformer encoder [49] as our basic block and try to make minimal changes. An encoder is a composition of two parts. The first part is constructed by a multi-head self-attention module and the second part is a feed-forward MLP with GELU non-linearity. The normalization layer is applied before both of the two parts. Both parts employ residual connection.

### 3.1  Memory-friendly Generator

The task of generation poses a high standard for spatial coherency in structure, color, and texture, both globally and locally. The transformer encoders take embedding token words as inputs and calculate the interaction between each token recursively. [63, 34]. The main dilemma here is: what is the right "word" for image generation tasks? If we similarly generate an image in a pixel-by-pixel manner through stacking transformer encoders, even a low-resolution image (e.g. $32 \times 32$) can result in an excessively long sequence (1024), causing the explosive cost of self-attention (quadratic w.r.t. the sequence length) and prohibiting the scalability to higher resolutions. To avoid this daunting cost, we are inspired by a common design philosophy in CNN-based GANs, to iteratively upscale the resolution at multiple stages [25, 16]. Our strategy is hence to increase the input sequence and reduce the embedding dimension gradually .

Figure 2 (left) illustrates a memory-friendly transformer-based generator that consists of multiple stages. Each stage stacks several transformer blocks. By stages, we gradually increase the feature

map resolution until it meets the target resolution $H \times W$. Specifically, the generator takes the random noise as its input, and passes it through a multiple-layer perceptron (MLP) to a vector of length $H_0 \times W_0 \times C$. The vector is reshaped into a $H_0 \times W_0$ resolution feature map (by default we use $H_0 = W_0 = 8$), each point a $C$-dimensional embedding. This "feature map" is next treated as a length-64 sequence of $C$-dimensional tokens, combined with the learnable positional encoding.

To scale up to higher-resolution images, we insert an upsampling module after each stage, consisting of a reshaping and resolution-upscaling layer. For lower-resolution stages (resolution lower than $64 \times 64$), the upsampling module firstly reshapes the 1D sequence of token embedding back to a 2D feature map $X_i \in \mathbb{R}^{H_i \times W_i \times C}$ and then adopts the `bicubic` layer to upsample its resolution while the embedded dimension is kept unchanged, resulting in the output $X_i^{'} \in \mathbb{R}^{2H_i \times 2W_i \times C}$. After that, the 2D feature map $X_i^{'}$ is again reshaped into the 1D sequence of embedding tokens. For higher-resolution stages, we replace the `bicubic` upscaling layer with the `pixelshuffle` module, which upsamples the resolution of feature map by $2\times$ ratio and also reduces the embedding dimension to a quarter of the input. This pyramid-structure with modified upscaling layers mitigates the memory and computation explosion. We repeat multiple stages until it reaches the target resolution $(H, W)$, and then we will project the embedding dimension to 3 and obtain the RGB image $Y \in \mathbb{R}^{H \times W \times 3}$.

### 3.2 Multi-scale Discriminator

Unlike the generator which synthesizes precise pixels, the discriminator is tasked to distinguish between real/fake images. This allows us to treat it as a typical classifier by simply tokenizing the input image in a coarser patch-level [34], where each patch can be regarded as a "word". However, compared to image recognition tasks where classifiers focus on the semantic differences, the discriminator executes a simpler and more detail-oriented task to distinguish between synthesized and real. Therefore, the local visual cues and artifacts will have an important effect on the discriminator. Practically, we observe that the patch splitting rule plays a crucial role, where large patch size sacrifices low-level texture details, and smaller patch size results in a longer sequence that costs more memory. The above dilemma motivates our design of multi-scale discriminator below.

As shown in Figure 2 (right), a multi-scale discriminator is designed to take varying size of patches as inputs, at its different stages. We firstly split the input images $Y \in R^{H \times W \times 3}$ into three different sequences by choosing different patch sizes $(P, 2P, 4P)$. The longest sequence $(\frac{H}{P} \times \frac{W}{P}) \times 3$ is linearly transformed to $(\frac{H}{P} \times \frac{W}{P}) \times \frac{C}{4}$ and then combined with the learnable position encoding to serve as the input of the first stage, where $\frac{C}{4}$ is the embedded dimension size. Similarly, the second and third sequences are linearly transformed to $(\frac{H}{2P} \times \frac{W}{2P}) \times \frac{C}{4}$ and $(\frac{H}{4P} \times \frac{W}{4P}) \times \frac{C}{2}$, and then separately concatenated into the second and third stages. Thus these three different sequences are able to extract both the semantic structure and texture details. Similar to the generator, we reshape the 1D-sentence to 2D feature map and adopt `Average Pooling` layer to downsample the feature map resolution, between each stage. By recursively forming the transformer blocks in each stage, we obtain a pyramid architecture where multi-scale representation is extracted. At the end of these blocks, a `[cls]` token is appended at the beginning of the 1D sequence and then taken by the classification head to output the real/fake prediction.

### 3.3 Grid Self-Attention: A Scalable Variant of Self-Attention for Image Generation

Self-attention allows the generator to capture the global correspondence, yet also impedes the efficiency when modeling long sequences/higher resolutions. That motivates many efficient self-attention designs in both language [64, 65] and vision tasks [66, 67]. To adapt self-attention for higher-resolution generative tasks, we propose a simple yet effective strategy, named *Grid Self-Attention*, tailored for high-resolution image generation.

As shown in Figure 3, instead of calculating the correspondence between a given token and all other tokens, the grid self-attention partitions the full-size feature map into several non-overlapped grids, and the token interactions are calculated inside each local grid. We add the grid self-attention on high-resolution stages (resolution higher than $32 \times 32$) while still keeping standard self-attention in low-resolution stages, shown as Figure 2, again so as to strategically balance local details and global awareness. The grid self-attention shows surprising effectiveness over other efficient self-attention forms [64, 67] in generative tasks, as compared later in Section 4.1.

One potential concern might arise with the boundary artifact between each grid. We observe that while the artifact indeed occurs at early training stages, it gradually vanishes given enough training

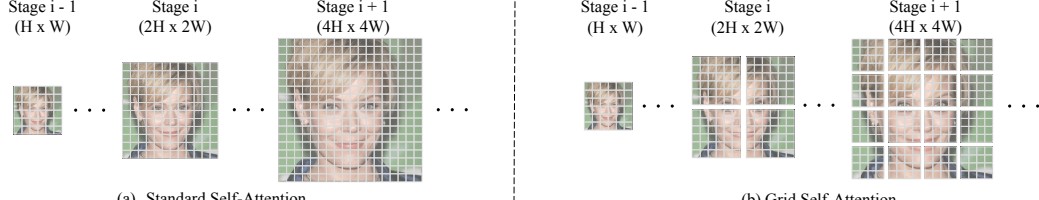

|   (a)   Standard Self-Attention   |   (b)   Grid Self-Attention   |

Figure 3: Grid Self-Attention across different transformer stages. We replace Standard Self-Attention with Grid Self-Attention when the resolution is higher than $32 \times 32$ and the grid size is set to be $16 \times 16$ by default.

iterations and training data, while producing nicely coherent final results. We think this is owing to the larger, multi-scale receptive field of the discriminator that requires generated image fidelity in different scales. For other cases where the large-scale training data is hard to obtain, we discuss several solutions on Sec. 4.6.

### 3.4   Exploring the Training Recipe

**Data Augmentation.** The transformer-based architectures are known to be highly data-hungry due to removing human-designed bias. Particularly in image recognition task [34], they were inferior to CNNs until much larger external data [68] was used for pre-training. To remove this roadblock, data augmentation was revealed as a blessing in [50], which showed that different types of strong data augmentation could lead us to data-efficient training for vision transformers.

We follow a similar mindset. Traditionally, training CNN-based GANs hardly refers to data augmentation. Recently, there is an interest surge in the few-shot GAN training, aiming to match state-of-the-art GAN results with orders of magnitude fewer real images [69, 70]. Contrary to this "commonsense" in CNNs, data augmentation is found to be crucial in transformer-based architectures, even with 100% real images being utilized. We show that simply using differential augmentation [69] with three basic operators $\{Translation, Cutout, Color\}$ leads to surprising performance improvement for TransGAN, while CNN-based GANs hardly benefit from it. We conduct a concrete study on the effectiveness of augmentation for both transformer and CNNs: see details in Section 4.2

**Relative Position Encoding.**   While classical transformers [49, 34] used deterministic position encoding or learnable position encoding, the relative position encoding [71] gains increasing popularity [72, 28, 52, 73], by exploiting lags instead of absolute positions. Considering a single head of self-attention layer,

$$Attention(Q, K, V) = softmax((\frac{QK^T}{\sqrt{d_k}}V) \tag{1}$$

where $Q$,$K$,$V \in \mathbb{R}^{(H \times W) \times C}$ represent query, key, value matrices, $H$,$W$,$C$ denotes the height, width, embedded dimension of the input feature map. The difference in coordinate between each query and key on $H$ axis lies in the range of $[-(H-1), H-1]$, and similar for $W$ axis. By simultaneously considering both $H$ and $W$ axis, the relative position can be represented by a parameterized matrix $M \in \mathbb{R}^{(2H-1) \times (2W-1)}$. Per coordinate, the relative position encoding $E$ is taken from matrix $M$ and added to the attention map $QK^T$ as a bias term, shown as following,

$$Attention(Q, K, V) = softmax(((\frac{QK^T}{\sqrt{d_k}} + E)V) \tag{2}$$

Compared to its absolute counterpart, relative position encoding learns a stronger "relationship" between local contents, bringing important performance gains in large-scale cases and enjoying widespread use ever since. We also observe it to consistently improve TransGAN, especially on higher-resolution datasets. We hence apply it on top of the learnable absolute positional encoding for both the generator and discriminator.

**Modified Normalization.** Normalization layers are known to help stabilize the deep learning training of deep neural networks, sometimes remarkably. While both the original transformer [49] and its variants [52, 54] by default use the layer normalization, we follow previous works [75, 16] and replace it with a token-wise scaling layer to prevent the magnitudes in transformer blocks from being too high, describe as $Y = X/\sqrt{\frac{1}{C}\sum_{i=0}^{C-1}(X^i)^2 + \epsilon}$, where $\epsilon = 1e-8$ by default, $X$ and $Y$ denote the token before and after scaling layer, $C$ represents the embedded dimension. Note that our modified normalization resembles local response normalization that was once used in AlexNet

Table 1: Unconditional image generation results on CIFAR-10, STl-10, and CelebA ($128 \times 128$) dataset. We train the models with their official code if the results are unavailable, denoted as "*", others are all reported from references.

| Methods | CIFAR-10 | | STL-10 | | CelebA |
|---|---|---|---|---|---|
| | IS↑ | FID↓ | IS↑ | FID↓ | FID↓ |
| WGAN-GP [1] | $6.49 \pm 0.09$ | 39.68 | - | - | - |
| SN-GAN [48] | $8.22 \pm 0.05$ | - | $9.16 \pm 0.12$ | 40.1 | - |
| AutoGAN [18] | $8.55 \pm 0.10$ | 12.42 | $9.16 \pm 0.12$ | 31.01 | - |
| AdversarialNAS-GAN [18] | $8.74 \pm 0.07$ | 10.87 | $9.63 \pm 0.19$ | 26.98 | - |
| Progressive-GAN [16] | $8.80 \pm 0.05$ | 15.52 | - | - | 7.30 |
| COCO-GAN [74] | - | - | - | - | 5.74 |
| StyleGAN-V2 [69] | 9.18 | 11.07 | $10.21^* \pm 0.14$ | 20.84* | 5.59* |
| StyleGAN-V2 + DiffAug. [69] | **9.40** | 9.89 | $10.31^* \pm 0.12$ | 19.15* | 5.40* |
| **TransGAN** | $9.02 \pm 0.12$ | **9.26** | **$10.43 \pm 0.16$** | **18.28** | **5.28** |

[75]. Unlike other "modern" normalization layers [76, 77, 78] that need affine parameters for both mean and variances, we find that a simple re-scaling without learnable parameters suffices to stabilize TransGAN training – in fact, it makes TransGAN train better and improves the FID.

## 4 Experiments

**Datasets** We start by evaluating our methods on three common testbeds: CIFAR-10 [79], STL-10 [80], and CelebA [81] dataset. The CIFAR-10 dataset consists of 60k $32 \times 32$ images, with 50k training and 10k testing images, respectively. We follow the standard setting to use the 50k training images without labels. For the STL-10 dataset, we use both the 5k training images and 100k unlabeled images, and all are resized to $48 \times 48$ resolution. For the CelebA dataset, we use 200k unlabeled face images (aligned and cropped version), with each image at $128 \times 128$ resolution. We further consider the CelebA-HQ and LSUN Church datasets to scale up TransGAN to higher resolution image generation tasks. We use 30k images for CelebA-HQ [16] dataset and 125k images for LSUN Church dataset [82], all at $256 \times 256$ resolution.

**Implementation** We follow the setting of WGAN [46], and use the WGAN-GP loss [1]. We adopt a learning rate of $1e - 4$ for both generator and discriminator, an Adam optimizer with $\beta_1 = 0$ and $\beta_2 = 0.99$, exponential moving average weights for generator, and a batch size of 128 for generator and 64 for discriminator, for all experiments. We choose DiffAug. [69] as basic augmentation strategy during the training process if not specially mentioned, and apply it to our competitors for a fair comparison. Other popular augmentation strategies ([70, 10]) are not discussed here since it is beyond the scope of this work. We use common evaluation metrics Inception Score (IS) [15] and Frechet Inception Distance (FID) [83], both are measured by 50K samples with their official Tensorflow implementations [1][2]. All experiments are set with 16 V100 GPUs, using PyTorch 1.7.0. We include detailed training cost for each dataset in Appendix D. We focus on the unconditional image generation setting for simplicity.

### 4.1 Comparison with State-of-the-art GANs

**CIFAR-10.** We compare TransGAN with recently published results by unconditional CNN-based GANs on the CIFAR-10 dataset, shown in Table 1. Note that some promising conditional GANs [21, 8] are not included, due to the different settings. As shown in Table 1, TransGAN surpasses the strong model of Progressive GAN [16], and many other latest competitors such as SN-GAN [48], AutoGAN [18], and AdversarialNAS-GAN [19], in terms of inception score (IS). It is only next to the huge and heavily engineered StyleGAN-v2 [40]. Once we look at the FID results, TransGAN is even found to outperform StyleGAN-v2 [40] with both applied the same data augmentation [69].

**STL-10.** We then apply TransGAN on another popular benchmark STL-10, which is larger in scale (105k) and higher in resolution (48x48). We compare TransGAN with both the automatic searched and hand-crafted CNN-based GANs, shown in Table 1. Different from the results on CIFAR-10, we find that TransGAN outperforms all current CNN-based GAN models, and sets **new state-of-the-art** results in terms of both IS and FID score. This is thanks to the fact that the STL-10 dataset size is $2\times$ larger than CIFAR-10, suggesting that transformer-based architectures benefit much more notably from larger-scale data than CNNs.

---

[1]https://github.com/openai/improved-gan/tree/master/inception_score
[2]https://github.com/bioinf-jku/TTUR

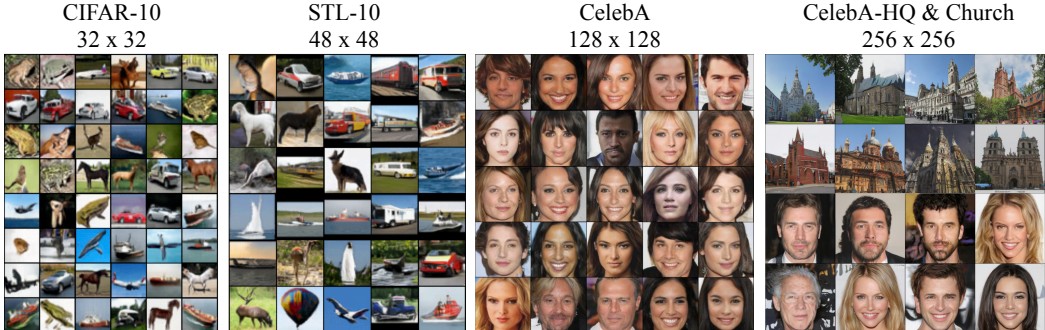

|  CIFAR-10
32 x 32 | STL-10
48 x 48 | CelebA
128 x 128 | CelebA-HQ & Church
256 x 256 |

Figure 4: Representative visual results produced by TransGAN on different datasets, as resolution grows from $32 \times 32$ to $256 \times 256$. More visual examples are included in Appendix F.

Table 2: The effectiveness of Data Augmentation on both CNN-based GANs and TransGAN. We use the full CIFAR-10 training set and DiffAug [69].

| Methods | WGAN-GP | | AutoGAN | | StyleGAN-V2 | | TransGAN | |
|---|---|---|---|---|---|---|---|---|
| | IS ↑ | FID ↓ | IS ↑ | FID ↓ | IS ↑ | FID ↓ | IS ↑ | FID ↓ |
| Original | **6.49** | 39.68 | 8.55 | **12.42** | 9.18 | 11.07 | 8.36 | 22.53 |
| + DiffAug [69] | 6.29 | **37.14** | **8.60** | 12.72 | **9.40** | **9.89** | **9.02** | **9.26** |

Table 3: The ablation study of proposed techniques in three common dataset CelebA($64 \times 64$), CelebA($128 \times 128$, and LSUN Church($256 \times 256$)). "OOM" represents out-of-momery issue.

| Training Configuration | CelebA
(64x64) | CelebA
(128x128) | LSUN Church
(256x256) |
|---|---|---|---|
| (A). Standard Self-Attention | 8.92 | **OOM** | **OOM** |
| (B). Nyström Self-Attention [64] | 13.47 | 17.42 | 39.92 |
| (C). Axis Self-Attention [67] | 12.39 | 13.95 | 29.30 |
| (D). Grid Self-Attention | 9.89 | 10.58 | 20.39 |
| + Multi-scale Discriminator | 9.28 | 8.03 | 15.29 |
| + Modified Normalization | 7.05 | 7.13 | 13.27 |
| + Relative Position Encoding | 6.14 | 6.32 | 11.93 |
| (E). Converge | **5.01** | **5.28** | **8.94** |

**CelebA (128x128).** We continue to examine another common benchmark: CelebA dataset ($128 \times 128$ resolution). As shown in Table 1, TransGAN largely outperforms Progressive-GAN [16] and COCO-GAN [74], and is slightly better than the strongest competitor StyleGAN-v2 [40], by reaching a FID score of 5.28. Visual examples generated on CIFAR-10, STL-10, and CelebA ($128 \times 128$) are shown in Figure 4, from which we observe pleasing visual details and diversity.

## 4.2 Scaling Up to Higher-Resolution

We further scale up TransGAN to higher-resolution ($256 \times 256$) generation, including on CelebA-HQ [16] and LSUN Church [82]. These high-resolution datasets are significantly more challenging due to their much richer and detailed low-level texture as well as the global composition. Thanks to the proposed multi-scale discriminator, TransGAN produces pleasing visual results, reaching competitive quantitative results with 10.28 FID on CelebA-HQ $256 \times 256$ and 8.94 FID on LSUN Church dataset, respectively. As shown in Figure 4, diverse examples with rich textures details are produced. We discuss the memory cost reduction brought by the Grid Self-Attention in Appendix E.

## 4.3 Data Augmentation is Crucial for TransGAN

We study the effectiveness of data augmentation for both CNN-based GANs and Our TransGAN. We apply the differentiable augmentation [69] to all these methods. As shown in Table 2, for three CNN-based GANs, the performance gains of data augmentation seems to diminish in the full-data regime. Only the largest model, StyleGAN-V2, is improved on both IS and FID. In sharp contrast, TransGAN sees a shockingly large margin of improvement: IS improving from 8.36 to 9.02 and FID improving from 22.53 to 9.26. This phenomenon suggests that CIFAR-10 is still "small-scale " when

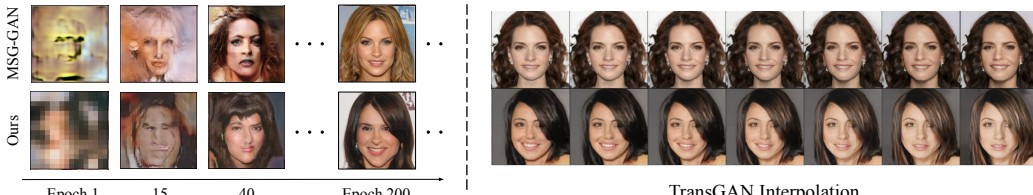

Figure 5: Left: training dynamic with training epochs for both TransGAN and MSG-GAN on CelebA-HQ ($256 \times 256$). Right: Interpolation on latent space produced by TransGAN.

fitting transformers; it re-confirms our assumption that transformer-based architectures are much more data-hungry than CNNs, and that can be helped by stronger data augmentation.

### 4.4 Ablation Study

To further evaluate the proposed grid self-attention, multi-scale discriminator, and unique training recipe, we conduct the ablation study by separately adding these techniques to the baseline method and report their FID score on different datasets. Due to the fact that most of our contributions are tailored for the challenges brought by higher-resolution tasks, we choose CelebA and LSUN Church as the main testbeds, with details shown in Table 3. We start by constructing our memory-friendly with vanilla discriminator as our baseline method (A), both applied with standard self-attention. The baseline method achieves relatively good results with 8.92 FID on CelebA ($64 \times 64$) dataset, however, it fail on higher-resolution tasks due to the memory explosion issue brought by self-attention. This motivates us to evaluate two efficient form of self-attention, (B) `Nyström Self-Attention` [64] and (C) `Axis Self-Attention` [67]

By replacing all self-attention layers in high-resolution stages (feature map resolution higher than $32 \times 32$) with these efficient variants, both two methods (B)(C) are able to produce reasonable results. However, they still show to be inferior to standard self-attention, even on the $64 \times 64$ resolution dataset. By adopting our proposed `Grid Self-Attention` (D), we observe a significant improvement on both three datasets, reaching 9.89, 10.58, 20.39 FID on CelebA $64 \times 64$, $128 \times 128$ and LSUN Church $256 \times 256$, respectively. Based on the configuration (D), we continue to add the proposed techniques, including the multi-scale discriminator, modified normalization, and relative position encoding. All these three techniques significantly improve the performance of TransGAN on three datasets. At the end, we train our final configuration (E) until it converges, resulting in the best FID on CelebA $64 \times 64$ (**5.01**), CelebA $128 \times 128$ (**5.28**), and LSUN Church $256 \times 256$ (**8.94**).

### 4.5 Understanding Transformer-based Generative Model

We dive deep into our transformer-based GAN by conducting interpolation on latent space and comparing its behavior with CNN-based GAN, through visualizing their training dynamics. We choose MSG-GAN [84] for comparison since it extracts multi-scale representation as well. As shown in Figure 5, the CNN-based GAN quickly extracts face representation in the early stage of training process while transformer only produces rough pixels with no meaningful global shape due to missing any inductive bias. However, given enough training iterations, TransGAN gradually learns informative position representation and is able to produce impressive visual examples at convergence. Meanwhile, the boundary artifact also vanishes at the end. For the latent space interpolation, TransGAN continues to show encouraging results where smooth interpolation are maintained on both local and global levels. More high-resolution visual examples will be presented in Appendix F.

### 4.6 Analyzing the Failure Cases and Improving High-resolution Tasks

While TransGAN shows competitive or even better results on common low-resolution benchmarks, we still see large improvement room of its performance on high-resolution synthesis tasks, by analyzing the failure cases shown in appendix C. Here we discuss several alternatives tailored for high-resolution synthesis tasks, as potential remedies to address these failure cases. Specifically, we apply the self-modulation [85, 22, 35] to our generator and use cross-attention [53, 86] to map the latent space to the global region. Besides, we replace the current $2\times$ upsampling layer, and instead firstly upsample it to $4\times$ lager resolution using bicubic interpolation, and then downsample it back to $2\times$ larger one. This simple modification not only helps the cross-boundary information interaction, but also help enhances the high-frequency details [87]. Moreover, an overlapped patch splitting strategy for discriminator can slightly improve the FID score. Additionally, we follow the

previous work [22, 40] to conduct noise injection before the self-attention layer, which is found to further improve the generation fidelity and diversity of TransGAN. By applying these techniques to our high-resolution GAN frameworks, we observe additional improvement on both qualitative and quantitative results, e.g., the FID score on CelebA $256 \times 256$ dataset is further improved from 10.26 to 8.93.

## 5 Conclusions, Limitation, and Discussions of Broad Impact

In this work, we provide the first pilot study of building GAN with pure transformers. We have carefully crafted the architectures and thoughtfully designed training techniques. As a result, the proposed TransGAN has achieved state-of-the-art performance across multiple popular datasets, and easily scales up to higher-resolution generative tasks. Although TransGAN provides an encouraging starting point, there is still a large room to explore further, such as achieving state-of-the-art results on $256 \times 256$ generation tasks or going towards extremely high resolution generation tasks (e.g., $1024 \times 1024$), which would be our future directions.

**Broader Impact.** The proposed generative model can serve as a data engine to alleviate the challenge of data collection. More importantly, using synthesized image examples helps avoid privacy concerns. However, the abuse of advanced generative models may create fake media materials, which demands caution in the future.

## Acknowledgements

We would like to express our deepest gratitude to the MIT-IBM Watson AI Lab, in particular John Cohn for generously providing us with the computing resources necessary to conduct this research. Z Wang's work is in part supported by an IBM Faculty Research Award, and the NSF AI Institute for Foundations of Machine Learning (IFML).

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
