# Appendix of "TransGAN: Two Pure Transformers Can Make One Strong GAN, and That Can Scale Up"

## A   Implementation of Data Augmentation

We mainly follow the way of differentiable augmentation to apply the data augmentation on our GAN training framework. Specifically, we conduct $\{Translation, Cutout, Color\}$ augmentation for TransGAN with probability $p$, while $p$ is empirically set to be $\{1.0, 0.3, 1.0\}$. However, we find that $Translation$ augmentation will hurt the performance of CNN-based GAN when 100% data is utilized. Therefore, we remove it and only conduct $\{Cutout, Color\}$ augmentation for AutoGAN. We also evaluate the effectiveness of stronger augmentation on high-resolution generative tasks (E.g. $256 \times 256$), including `random-cropping`, `random hue adjustment`, and `image filtering`. Moreover, we find `image filtering` helps remove the boundary artifacts in a very early stage of training process, while it takes longer training iterations to remove it in the original setting.

## B   Detailed Architecture Configurations

We present the specific architecture configurations of TransGAN on different datasets, shown in Table 1, 2, 3, 4. For the generator architectures, the "Block" represents the basic Transformer Block constructed by self-atention, Normalization, and Feed-forward MLP. "Grid Block" denotes the Transformer Block where the standard self-attention is replaced by the propose `Grid Self-Attention`, with grid size equals to 16. `Upsampling` layer represents `Bicubic Upsampling` by default. The "input_shape" and "output_shape" denotes the shape of input feature map and output feature map, respectively. For the discriminator architectures, we use "Layer Flatten" to represent the process of patch splitting and linear transformation. In each stage, the output feature map is concatenated with another different sequence, as described in Sec. 3.2. In the final stage, we add another CLS token and use a Transformer Block to build correspondence between CLS token and extracted representation. In the end, only the CLS token is taken by the Classification Head for predicting real/fake. For low-resolution generative tasks (e.g., CIFAR-10 and STL-10), we only split the input images into two different sequences rather than three and only two stages are built as well.

## C   Failure Cases Analysis

Since TransGAN shows inferior FID scores compared to state-of-the-art ConvNet-based GAN on high-resolution synthesis tasks, we try to visualize the failure cases of TransGAN on CelebA-HQ $256 \times 256$ dataset, to better understand its drawbacks. As shown in Fig. 1, We pick several representative failure examples produced by TransGAN. We observe that most failure examples are from the "wearing glasses" class and side faces, which indicates that TransGAN may also suffer from the imbalanced data distribution issue, as well as the issue of insufficient training data. We believe this could be also a very interesting question and will explore it further in the near future.

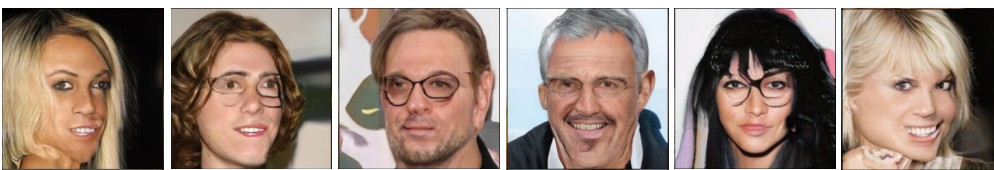

Figure 1: Analyzing the failure cases produced by TransGAN on High-resolution synthesis tasks.

## D   Training Cost

We include the training cost of TransGAN on different datasets, with resolutions across from $32 \times 32$ to $256 \times 256$, shown in Table 5. The largest experiment costs around 3 days with 32 V100 GPUs.

35th Conference on Neural Information Processing Systems (NeurIPS 2021), Sydney, Australia.

Table 1: Architecture configuration of TransGAN on CIFAR-10 dataset.

**Generator**

| Stage | Layer | Input Shape | Output Shape |
|---|---|---|---|
| - | MLP | $512$ | $(8 \times 8) \times 1024$ |
| 1 | Block | $(8 \times 8) \times 1024$ | $(8 \times 8) \times 1024$ |
| | Block | $(8 \times 8) \times 1024$ | $(8 \times 8) \times 1024$ |
| | Block | $(8 \times 8) \times 1024$ | $(8 \times 8) \times 1024$ |
| | Block | $(8 \times 8) \times 1024$ | $(8 \times 8) \times 1024$ |
| | Block | $(8 \times 8) \times 1024$ | $(8 \times 8) \times 1024$ |
| 2 | PixelShuffle | $(8 \times 8) \times 1024$ | $(16 \times 16) \times 256$ |
| | Block | $(16 \times 16) \times 256$ | $(16 \times 16) \times 256$ |
| | Block | $(16 \times 16) \times 256$ | $(16 \times 16) \times 256$ |
| | Block | $(16 \times 16) \times 256$ | $(16 \times 16) \times 256$ |
| | Block | $(16 \times 16) \times 256$ | $(16 \times 16) \times 256$ |
| 3 | PixelShuffle | $(16 \times 16) \times 256$ | $(32 \times 32) \times 64$ |
| | Block | $(32 \times 32) \times 64$ | $(32 \times 32) \times 64$ |
| | Block | $(32 \times 32) \times 64$ | $(32 \times 32) \times 64$ |
| - | Linear Layer | $(32 \times 32) \times 64$ | $32 \times 32 \times 3$ |

**Discriminator**

| Stage | Layer | Input Shape | Out Shape |
|---|---|---|---|
| - | Linear Flatten | $32 \times 32 \times 3$ | $(16 \times 16) \times 192$ |
| 1 | Block | $(16 \times 16) \times 192$ | $(16 \times 16) \times 192$ |
| | Block | $(16 \times 16) \times 192$ | $(16 \times 16) \times 192$ |
| | Block | $(16 \times 16) \times 192$ | $(16 \times 16) \times 192$ |
| | AvgPooling | $(16 \times 16) \times 192$ | $(8 \times 8) \times 192$ |
| | Concatenate | $(8 \times 8) \times 192$ | $(8 \times 8) \times 384$ |
| 2 | Block | $(8 \times 8) \times 384$ | $(8 \times 8) \times 384$ |
| | Block | $(8 \times 8) \times 384$ | $(8 \times 8) \times 384$ |
| | Block | $(8 \times 8) \times 384$ | $(8 \times 8) \times 384$ |
| - | Add CLS Token | $(8 \times 8) \times 384$ | $(8 \times 8 + 1) \times 384$ |
| | Block | $(8 \times 8 + 1) \times 384$ | $(8 \times 8 + 1) \times 384$ |
| | CLS Head | $1 \times 384$ | $1$ |

Table 2: Architecture configuration of TransGAN on STL-10 dataset.

**Generator**

| Stage | Layer | Input Shape | Output Shape |
|---|---|---|---|
| - | MLP | $512$ | $(12 \times 12) \times 1024$ |
| 1 | Block | $(12 \times 12) \times 1024$ | $(12 \times 12) \times 1024$ |
| | Block | $(12 \times 12) \times 1024$ | $(12 \times 12) \times 1024$ |
| | Block | $(12 \times 12) \times 1024$ | $(12 \times 12) \times 1024$ |
| | Block | $(12 \times 12) \times 1024$ | $(12 \times 12) \times 1024$ |
| | Block | $(12 \times 12) \times 1024$ | $(12 \times 12) \times 1024$ |
| 2 | PixelShuffle | $(12 \times 12) \times 1024$ | $(24 \times 24) \times 256$ |
| | Block | $(24 \times 24) \times 256$ | $(24 \times 24) \times 256$ |
| | Block | $(24 \times 24) \times 256$ | $(24 \times 24) \times 256$ |
| | Block | $(24 \times 24) \times 256$ | $(24 \times 24) \times 256$ |
| | Block | $(24 \times 24) \times 256$ | $(24 \times 24) \times 256$ |
| 3 | PixelShuffle | $(24 \times 24) \times 256$ | $(48 \times 48) \times 64$ |
| | Block | $(48 \times 48) \times 64$ | $(48 \times 48) \times 64$ |
| | Block | $(48 \times 48) \times 64$ | $(48 \times 48) \times 64$ |
| - | Linear Layer | $(48 \times 48) \times 64$ | $48 \times 48 \times 3$ |

**Discriminator**

| Stage | Layer | Input Shape | Out Shape |
|---|---|---|---|
| - | Linear Flatten | $48 \times 48 \times 3$ | $(16 \times 16) \times 192$ |
| 1 | Block | $(24 \times 24) \times 192$ | $(24 \times 24) \times 192$ |
| | Block | $(24 \times 24) \times 192$ | $(24 \times 24) \times 192$ |
| | Block | $(24 \times 24) \times 192$ | $(24 \times 24) \times 192$ |
| | AvgPooling | $(24 \times 24) \times 192$ | $(12 \times 12) \times 192$ |
| | Concatenate | $(12 \times 12) \times 192$ | $(12 \times 12) \times 384$ |
| 2 | Block | $(12 \times 12) \times 384$ | $(12 \times 12) \times 384$ |
| | Block | $(12 \times 12) \times 384$ | $(12 \times 12) \times 384$ |
| | Block | $(12 \times 12) \times 384$ | $(12 \times 12) \times 384$ |
| - | Add CLS Token | $(12 \times 12) \times 384$ | $(12 \times 12 + 1) \times 384$ |
| | Block | $(12 \times 12 + 1) \times 384$ | $(12 \times 12 + 1) \times 384$ |
| | CLS Head | $1 \times 384$ | $1$ |

# E   Memory Cost Comparison

We compare the GPU memory cost between standard self-attention and grid self-attention. Our testbed is set on Nvidia V100 GPU with batch size set to 1, using Pytorch V1.7 environment. We evaluate the inference cost of these two architectures, without calculating the gradient. Since the original self-attention will cause out-of-memory issue even when batch size is set to 1, we reduce the model size on $(256 \times 256)$ resolution tasks to make it fit GPU memory, and apply the same strategy on $128 \times 128$ and $64 \times 64$ architectures as well. When evaluating the grid self-attention, we do not reduce the model size and only modify the standard self-attention on the specific stages where the resolution is larger than $32 \times 32$, and replace it with the proposed Grid Self-Attention. As shown in in Figure 2, even the model size of the one that represents the standard self-attention is reduced, it still costs significantly larger GPU memory than the proposed Grid Self-Attention does.

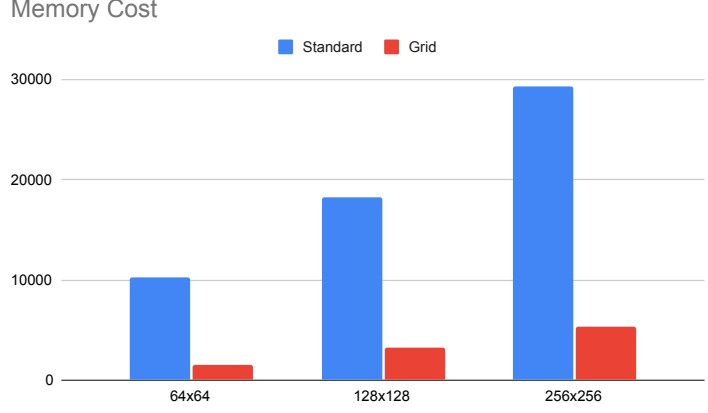

Figure 2: Memory cost comparison between standard self-attention and grid self-attention

Table 3: Architecture configuration of TransGAN on CelebA ($128 \times 128$) dataset.

**Generator**

| Stage | Layer | Input Shape | Output Shape |
|---|---|---|---|
| - | MLP | 512 | $(8 \times 8) \times 1024$ |
| 1 | Block | $(8 \times 8) \times 1024$ | $(8 \times 8) \times 1024$ |
| | Block | $(8 \times 8) \times 1024$ | $(8 \times 8) \times 1024$ |
| | Block | $(8 \times 8) \times 1024$ | $(8 \times 8) \times 1024$ |
| | Block | $(8 \times 8) \times 1024$ | $(8 \times 8) \times 1024$ |
| | Block | $(8 \times 8) \times 1024$ | $(8 \times 8) \times 1024$ |
| 2 | Upsampling | $(8 \times 8) \times 1024$ | $(16 \times 16) \times 1024$ |
| | Block | $(16 \times 16) \times 1024$ | $(16 \times 16) \times 1024$ |
| | Block | $(16 \times 16) \times 1024$ | $(16 \times 16) \times 1024$ |
| | Block | $(16 \times 16) \times 1024$ | $(16 \times 16) \times 1024$ |
| | Block | $(16 \times 16) \times 1024$ | $(16 \times 16) \times 1024$ |
| 3 | PixelShuffle | $(16 \times 16) \times 1024$ | $(32 \times 32) \times 256$ |
| | Block | $(32 \times 32) \times 256$ | $(32 \times 32) \times 256$ |
| | Block | $(32 \times 32) \times 256$ | $(32 \times 32) \times 256$ |
| | Block | $(32 \times 32) \times 256$ | $(32 \times 32) \times 256$ |
| | Block | $(32 \times 32) \times 256$ | $(32 \times 32) \times 256$ |
| 4 | PixelShuffle | $(32 \times 32) \times 256$ | $(64 \times 64) \times 64$ |
| | Grid Block | $(64 \times 64) \times 64$ | $(64 \times 64) \times 64$ |
| | Grid Block | $(64 \times 64) \times 64$ | $(64 \times 64) \times 64$ |
| | Grid Block | $(64 \times 64) \times 64$ | $(64 \times 64) \times 64$ |
| | Grid Block | $(64 \times 64) \times 64$ | $(64 \times 64) \times 64$ |
| 5 | PixelShuffle | $(64 \times 64) \times 64$ | $(128 \times 128) \times 16$ |
| | Grid Block | $(128 \times 128) \times 16$ | $(128 \times 128) \times 16$ |
| | Grid Block | $(128 \times 128) \times 16$ | $(128 \times 128) \times 16$ |
| | Grid Block | $(128 \times 128) \times 16$ | $(128 \times 128) \times 16$ |
| | Grid Block | $(128 \times 128) \times 16$ | $(128 \times 128) \times 16$ |
| - | Linear Layer | $(128 \times 128) \times 16$ | $128 \times 128 \times 3$ |

**Discriminator**

| Stage | Layer | Input Shape | Out Shape |
|---|---|---|---|
| - | Linear Flatten | $128 \times 128 \times 3$ | $(32 \times 32) \times 96$ |
| 1 | Block | $(32 \times 32) \times 96$ | $(32 \times 32) \times 96$ |
| | Block | $(32 \times 32) \times 96$ | $(32 \times 32) \times 96$ |
| | Block | $(32 \times 32) \times 96$ | $(32 \times 32) \times 96$ |
| | AvgPooling | $(32 \times 32) \times 96$ | $(16 \times 16) \times 96$ |
| | Concatenate | $(16 \times 16) \times 96$ | $(16 \times 16) \times 192$ |
| 2 | Block | $(16 \times 16) \times 192$ | $(16 \times 16) \times 192$ |
| | Block | $(16 \times 16) \times 192$ | $(16 \times 16) \times 192$ |
| | Block | $(16 \times 16) \times 192$ | $(16 \times 16) \times 192$ |
| | AvgPooling | $(16 \times 16) \times 192$ | $(8 \times 8) \times 192$ |
| | Concatenate | $(8 \times 8) \times 192$ | $(8 \times 8) \times 384$ |
| 3 | Block | $(8 \times 8) \times 192$ | $(8 \times 8) \times 384$ |
| | Block | $(8 \times 8) \times 384$ | $(8 \times 8) \times 384$ |
| | Block | $(8 \times 8) \times 384$ | $(8 \times 8) \times 384$ |
| - | Add CLS Token | $(8 \times 8) \times 384$ | $(8 \times 8 + 1) \times 384$ |
| | Block | $(8 \times 8 + 1) \times 384$ | $(8 \times 8 + 1) \times 384$ |
| | CLS Head | $1 \times 384$ | 1 |

Table 4: Architecture configuration of TransGAN on CelebA ($256 \times 256$) and LSUN Church ($256 \times 256$) dataset.

**Generator**

| Stage | Layer | Input Shape | Output Shape |
|---|---|---|---|
| - | MLP | 512 | $(8 \times 8) \times 1024$ |
| 1 | Block | $(8 \times 8) \times 1024$ | $(8 \times 8) \times 1024$ |
| | Block | $(8 \times 8) \times 1024$ | $(8 \times 8) \times 1024$ |
| | Block | $(8 \times 8) \times 1024$ | $(8 \times 8) \times 1024$ |
| | Block | $(8 \times 8) \times 1024$ | $(8 \times 8) \times 1024$ |
| | Block | $(8 \times 8) \times 1024$ | $(8 \times 8) \times 1024$ |
| 2 | Upsampling | $(8 \times 8) \times 1024$ | $(16 \times 16) \times 1024$ |
| | Block | $(16 \times 16) \times 1024$ | $(16 \times 16) \times 1024$ |
| | Block | $(16 \times 16) \times 1024$ | $(16 \times 16) \times 1024$ |
| | Block | $(16 \times 16) \times 1024$ | $(16 \times 16) \times 1024$ |
| | Block | $(16 \times 16) \times 1024$ | $(16 \times 16) \times 1024$ |
| 3 | Upsampling | $(16 \times 16) \times 1024$ | $(32 \times 32) \times 1024$ |
| | Block | $(32 \times 32) \times 1024$ | $(32 \times 32) \times 1024$ |
| | Block | $(32 \times 32) \times 1024$ | $(32 \times 32) \times 1024$ |
| | Block | $(32 \times 32) \times 1024$ | $(32 \times 32) \times 1024$ |
| | Block | $(32 \times 32) \times 1024$ | $(32 \times 32) \times 1024$ |
| 4 | PixelShuffle | $(32 \times 32) \times 1024$ | $(64 \times 64) \times 256$ |
| | Grid Block | $(64 \times 64) \times 256$ | $(64 \times 64) \times 256$ |
| | Grid Block | $(64 \times 64) \times 256$ | $(64 \times 64) \times 256$ |
| | Grid Block | $(64 \times 64) \times 256$ | $(64 \times 64) \times 256$ |
| | Grid Block | $(64 \times 64) \times 256$ | $(64 \times 64) \times 256$ |
| 5 | PixelShuffle | $(64 \times 64) \times 256$ | $(128 \times 128) \times 64$ |
| | Grid Block | $(128 \times 128) \times 64$ | $(128 \times 128) \times 64$ |
| | Grid Block | $(128 \times 128) \times 64$ | $(128 \times 128) \times 64$ |
| | Grid Block | $(128 \times 128) \times 64$ | $(128 \times 128) \times 64$ |
| | Grid Block | $(128 \times 128) \times 64$ | $(128 \times 128) \times 64$ |
| 6 | PixelShuffle | $(128 \times 128) \times 64$ | $(256 \times 256) \times 16$ |
| | Grid Block | $(256 \times 256) \times 16$ | $(256 \times 256) \times 16$ |
| | Grid Block | $(256 \times 256) \times 16$ | $(256 \times 256) \times 16$ |
| | Grid Block | $(256 \times 256) \times 16$ | $(256 \times 256) \times 16$ |
| | Grid Block | $(256 \times 256) \times 16$ | $(256 \times 256) \times 16$ |
| - | Linear Layer | $(256 \times 256) \times 16$ | $256 \times 256 \times 3$ |

**Discriminator**

| Stage | Layer | Input Shape | Out Shape |
|---|---|---|---|
| - | Linear Flatten | $256 \times 256 \times 3$ | $(64 \times 64) \times 96$ |
| 1 | Block | $(64 \times 64) \times 96$ | $(64 \times 64) \times 96$ |
| | Block | $(64 \times 64) \times 96$ | $(64 \times 64) \times 96$ |
| | Grid Block | $(64 \times 64) \times 96$ | $(64 \times 64) \times 96$ |
| | AvgPooling | $(64 \times 64) \times 96$ | $(32 \times 32) \times 96$ |
| | Concatenate | $(32 \times 32) \times 96$ | $(32 \times 32) \times 192$ |
| 2 | Block | $(32 \times 32) \times 192$ | $(32 \times 32) \times 192$ |
| | Block | $(32 \times 32) \times 192$ | $(32 \times 32) \times 192$ |
| | Block | $(32 \times 32) \times 192$ | $(32 \times 32) \times 192$ |
| | AvgPooling | $(32 \times 32) \times 192$ | $(16 \times 16) \times 192$ |
| | Concatenate | $(16 \times 16) \times 192$ | $(16 \times 16) \times 384$ |
| 3 | Block | $(16 \times 16) \times 192$ | $(16 \times 16) \times 384$ |
| | Block | $(16 \times 16) \times 384$ | $(16 \times 16) \times 384$ |
| | Block | $(16 \times 16) \times 384$ | $(16 \times 16) \times 384$ |
| - | Add CLS Token | $(16 \times 16) \times 384$ | $(16 \times 16 + 1) \times 384$ |
| | Block | $(16 \times 16 + 1) \times 384$ | $(16 \times 16 + 1) \times 384$ |
| | CLS Head | $1 \times 384$ | 1 |

# F   Visual Examples

We include more high-resolution visual examples on Figure 3,4. The visual examples produced by TransGAN show impressive details and diversity.

Table 5: Training Configuration

| Dataset | Size | Resolution | GPUs | Epochs | Time |
|---------|------|------------|------|--------|------|
| CIFAR-10 | 50k | $32 \times 32$ | 2 | 500 | 2.6 days |
| STL-10 | 105k | $48 \times 48$ | 4 | 200 | 2.0 days |
| CelebA | 200k | $64 \times 64$ | 8 | 250 | 2.4 days |
| CelebA | 200k | $128 \times 128$ | 16 | 250 | 2.1 days |
| CelebA-HQ | 30k | $256 \times 256$ | 32 | 300 | 2.9 days |
| LSUN Church | 125k | $256 \times 256$ | 32 | 120 | 3.2 days |

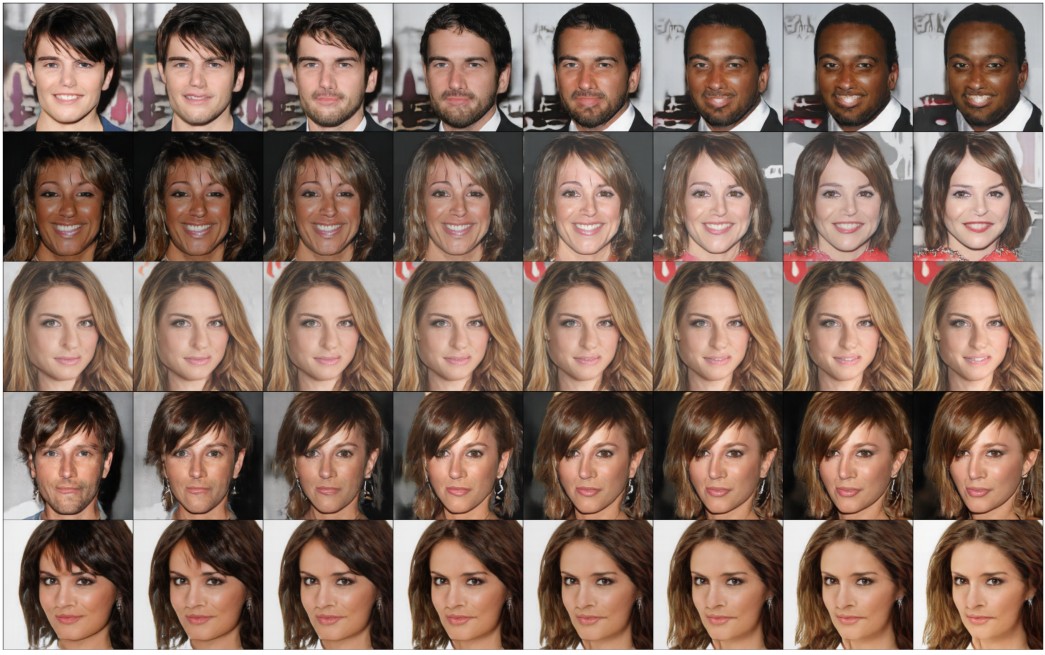

Figure 3: Latent Space Interpolation on CelebA ($256 \times 256$) dataset.

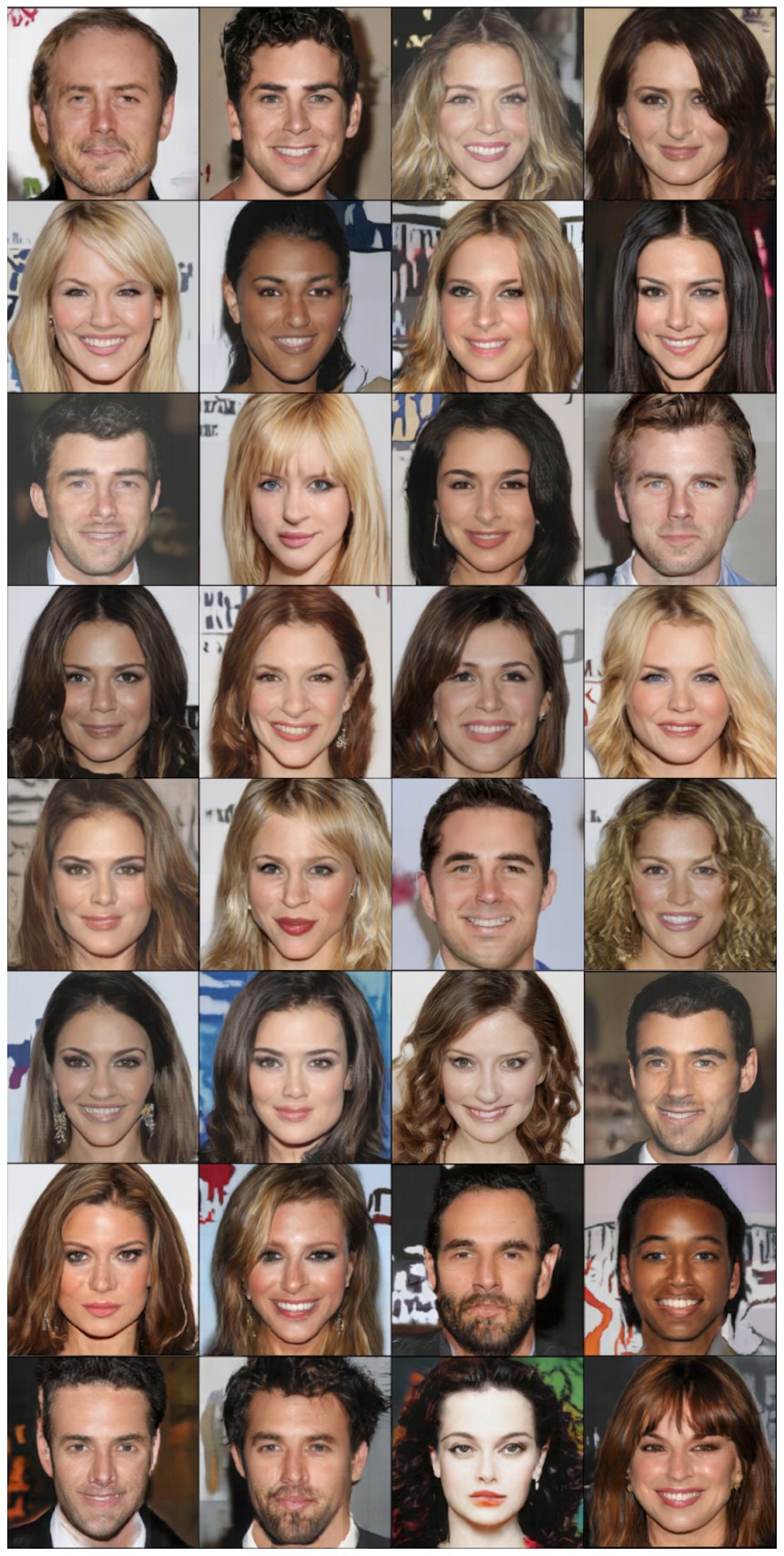

Figure 4: High-resolution representative visual examples on CelebA ($256 \times 256$) dataset.