# OpenReview forum: "TransGAN: Two Pure Transformers Can Make One Strong GAN, and That Can Scale Up"
_NeurIPS.cc/2021/Conference — NeurIPS 2021 Poster_

### Official Review · Reviewer_PbKe · 2021-07-13

**Rating:** 8
**Confidence:** 4

**Summary:**

This paper proposes to build a GAN model using only transformer blocks, and completely free of convolutions. Several architecture designs are proposed and a unique training recipe is also introduced for stably training the proposed TransGAN. TransGAN  can scale up to high resolution (e.g. 256 × 256), and delivers state-of-the-art results on multiple benchmarks.

**Limitations And Societal Impact:**

The social impact has been discussed by the authors at the end of the paper.

**Main Review:**

Strengths:
+ TransGAN is the first attempt to use only transformers to build a strong GAN. While the backbone’s building blocks is not such novel, the target task and the way to tackle it is very new and deserves to be encouraged.
+ The authors have extensively introduce the architectures of the discriminator, the self-attention mechanism and the generator. They have revealed useful design insights for transformer-based image generation.
+ The authors investigate many techniques to stabilize the network training and improve the performance, including data augmentation, relative position encoding, and simpler normalization borrowed from [73, 16].
+ The proposed method shows strong empirical results as well as scalability. TransGAN achieves SOTA results on STL-10 and CelebA-128, is close to SOTA and outperforms StyleGAN-v2 (even with DiffAug) on CIFAR-10, and remains very competitive on CelebA-256 and LSUN-Church 256.
+ The visual quality of generated results at 256 level is impressive and can rival the best CNN GANs. This is one big step forward towards justifying transformer-based generation.
+ Ablation experiments on each architectures or training algorithm components are very thorough and convincing.
+ Figure 5 shows intermediate generation results to compare TransGAN behavior with CNN-based GAN, as well as interpolation on latent space. They looks very interesting: TransGAN learns to generate images with global awareness, totally different from the local refinement regime of CNN GANs; meanwhile, the latent space seems the same smooth.
+ The paper is overall quite well-written and easy to follow.

Weaknesses:
- Corresponding to the first strength, I have to point out that no building block in TransGAN is particularly novel. However, I actually feel this fine and acceptable, as it demonstrates the value of putting each moving part together in one right pipeline.
- The description of Multi-scale Discriminator is complicated and I’m not sure if I followed well. Is this anything similar to token-to-token ViT, or Pyramid ViT? Since the idea of progressively changing tokens has been exploited by several ViT works, the authors need to describe their context better.
- Also for the grid self-attention, please discuss its relationship to Swin Transformer and others window-based ViTs.
- Why WGAN-GP loss is chosen? Is gradient penalty your necessary training component? Can other losses (e.g., Hinge, least-square) work with TransGAN?
- Please report TransGAN model size and/or memory usage.
- Minor comments: This paper makes big progress to high resolution generation task, but there is still a gap behind the highest-resolution that CNN GANs can now scale up to, e.g. 1024. While it can reasonably go to the authors’ future work, some discussions to shed light on how they plan to do so would be welcomed.


**Time Spent Reviewing:**

3

---

> ### Author Response · Authors · 2021-08-10
> **Response to Reviewer PbKe**
>
> **Q1**: Since the idea of progressively changing tokens has been exploited by several ViT works, the authors need to describe their context better.
> **A**: The idea of multi-scale discriminator shares similar design philosophy with Token-to-Token and PVT, since all of them try to save computational cost and extract multi-scale representations. However, ours extracts multi-scale representation by directly splitting different patches on input images, which preserves more information from inputs. Meanwhile, the multi-scale discriminator is also equipped with the proposed grid self-attention, which makes it more scalable to higher resolution tasks.
>
>
> **Q2**: Relationship between Grid self-attention and Swin Transformer.
> **A**: Both the grid self-attention and swin transformer aim for the scalability of vision transformers. The swin transformer proposes the shifted windows to capture local relationships. However, the “shift” feature makes its practical implementation much more complex than the grid self-attention. Moreover, TransGAN only employs grid self-attention on the stages when resolution is larger than 32x32, thus it can capture both the long- and short-range dependencies simultaneously. We do not observe obvious improvement when applying the “shift” feature in our architectures.
>
>
>
> **Q3**: Why WGAN-GP loss is chosen? Is gradient penalty your necessary training component? Can other losses (e.g., Hinge, least-square) work with TransGAN?
> **A**: We follow the previous setting [16, 82] to choose WGAN-GP loss and observe that gradient penalty can provide significant improvement since it helps the lipschitz continuity. Other losses (like Hinge loss and Least-square loss) are mostly adopted when the discriminator is equipped with spectral normalization [46], to maintain the lipschitz continuity. However, our main goal is to keep the original transformer block unchanged with the fewest possible modifications and explore its potential power. Adding spectral normalization [46] into the transformer discriminator will be a very good way to explore further, which is currently beyond the scope of this work. We will add more discussion on those loss functions in our final manuscript.
>
>
> **Q4**: Please report TransGAN model size and/or memory usage.
> **A**:  We report the FLOPs cost, model size, and memory cost of TransGAN on different resolutions. Our testbed is set on Nvidia V100 GPU with batch size set to 1, using Pytorch V1.7 environment. We evaluate the inference cost of our generator, without calculating the gradient.
>
> | Resolution | FLOPS/G | Params/M | Memory/M |
> |:----------:|:-------:|:--------:|:--------:|
> |   32 x 32  |  2.787  |   49.88  |    329   |
> |  128 x 128 |  19.29  |   79.28  |   2012   |
> |  256 x 256 |  48.04  |  119.18  |   5203   |
>
> **Q5**: Some discussions to shed light on how authors plan to do 1024 resolution would be welcomed.
> **A**: As the current design (multi-scale discriminator and grid self-attention) allows the scalability of TransGAN architecture, it will be much easier to target at higher resolution tasks (e.g., 1024), compared to the vanilla transformer architecture. Besides, the stronger data augmentation (e.g. filtering, or geometric transformation ) and regularization are also potential tools to make TransGAN stronger. We will explore it in our future work.

---

> > ### Comment · Reviewer_PbKe · 2021-08-20
> > **Post rebuttal update**
> >
> > Thanks for the response. I have read the authors' rebuttal and other reviewers' comments. The rebuttal has addressed most of my concerns and I will keep my initial rating.

---

### Official Review · Reviewer_M68S · 2021-07-16

**Rating:** 6
**Confidence:** 5

**Summary:**


This work proposes TransGAN by replacing the CNN-based structure of both the generator and the discriminator in GANs with Transformer-based structure. To reduce the computing load, grid self-attention is proposed. Meanwhile, a relative position encoding scheme is introduced to the attention module to make the model better aware of the position information. Experimental results suggest that TransGAN achieves comparable performance to state-of-the-art CNN-based GANs.

**Limitations And Societal Impact:**

- Why not reporting FID on CelebA-HQ 256 and Church 256?
- Since the improvement of TransGAN over StyleGAN2 is marginal (or even worse), what is the advantage of using Transformer instead of CNN? The authors claim that "convolutions have strong inductive bias" (Line 46), but do not show that why transformer can solve this problem?
- The interpolation is not that smooth as claimed by the authors. Please see the hair region of Fig. 1(b).
- An important ablation is missing. The title is "two" transformers make one strong GAN. What if we only use one transformer? The authors only compare TransGAN (G-trans, D-trans) with state-of-the-art GANs (G-CNN, D-CNN), without the comparison to (G-CNN, D-trans) and (G-trans and D-CNN). In this way, I am not sure on which part the transformer-based structure is helpful.


**Main Review:**


- Studying the GAN architecture is encouraging.
- Grid self-attention makes it possible to apply transformer to *large-scale* GAN training.

**Time Spent Reviewing:**

3 hours

---

> ### Author Response · Authors · 2021-08-10
> **Response to Reviewer M68S**
>
> **Q1**: Why not report FID on CelebA-HQ 256 and Church 256?
> **A**: The FID results are reported in Line 300, where TransGAN achieves 10.28 (CelebA-HQ) and 8.94 (Church), respectively. Although the quantitative scores on higher resolution tasks are slightly inferior to state-of-the-art ConvNet-based GANs, their achieved visual qualities are recognized as comparable. More importantly, SOTA ConvNet-based GANs have been explored for years and empowered by numerous clever training tricks that are orthogonal to architecture innovations. TransGAN has not fully exploited such wealth yet, and hence it has room for improvement here.
>
> Take StyleGAN-v2 as an example, it employs noise injection, equalized learning rates, weight demodulation, lazy regularization, and mini-batch standard deviation, all contributing to stabilizing its training process and attaining the final fine generation quality. During rebuttal week, we tried one of its tricks, the noise injection [22, 40], on the inputs of all self-attention modules in the TransGAN generator. We are delighted to find it to immediately boost TransGAN FID results, e.g., from 8.94 to 7.48 on LSUN-Church (256x256). We are now continuing to try other training techniques on TransGAN, and expect to see them to bring more FID boosts.
>
> **Q3**: The interpolation is not that smooth as claimed by the authors. Please see the hair region of Fig. 1(b).
> **A**: The interpolation results are randomly sampled from the results produced by TransGAN. We agree there are some subtle artifacts, but point out the overall quality remains consistently good for all sampled images. We promise to include more discussion as well as failure cases in our final draft. Besides, we believe this observation may also motivate the usage of smooth optimizers [a, b] for TransGAN, which will be our future work.
>
>
> **Q4**: Since the improvement of TransGAN over StyleGAN2 is marginal (or even worse), what is the advantage of using Transformer instead of CNN?
> **A**: The goal of our work is not to pursue a state-of-the-art GAN approach or to replace StyleGAN-v2 with another backbone. Instead, this paper aims to conduct the first pilot study to build a GAN completely free of convolutions, using only pure transformer-based architectures. That said, “marginal improvement or not” is perhaps not a fair judgement of this work’s novelty or impact. We view the most exciting innovation of this work as follows:
> 1) TransGAN is the first attempt to use only transformers to build a strong GAN. We present a completely new pipeline towards building the pure transformer-based GAN architectures, where a coherent set of techniques and innovations are proposed to make it stable and scalable.
> 2) Our work challenges a “commonsense” in GAN architecture design, where convolution is necessary for building a state-of-the-art GAN. We provide solid experiments to show the effectiveness of pure transformer-based architecture.
> 3) We push the limits of the vision transformer on generative tasks and explore its potential to become the “universal” architecture among different tasks. Our encouraging results may potentially motivate more interesting works, such as text-to-image generation tasks and so on. As pointed out by Reviewer c687: “this work will likely influence future works in generative modeling”.
>
> **Q5**: The authors claim that "convolutions have strong inductive bias" (Line 46), but do not show why the transformer can solve this problem?
> **A**: The strong inductive bias is claimed more like a feature rather than a problem of convolution,  as illustrated in Sec. 1. The goal of this work is not exactly to “solve this problem”, yet to examine convolution’s necessity in GAN training and see if we could approach image generation from the other perspective. Since convolutions are now known to have simplified modeling of visual signals (e.g., the local receptive field and spatial invariance), transformers in contrast have “global” views and can apply spatially varying transformations to extract heterogeneous regional patterns. The previous open question was whether this highly flexible form can be tamed to model and generate the highly structured (and still, mostly smooth) images. This paper provides the first positive answer: if transformers can be trained with sufficient data, stronger data augmentations, et. al., they can indeed outperform SOTA ConvNet competitors. In other words, the success of TransGAN provides new reflections on the pros and cons, as well as the necessity of using convolution for image generations, bringing in new inspirations and angles.
>
>
>
> **Q6**: What if we only use one transformer? The authors only compare TransGAN (G-trans, D-trans) with state-of-the-art GANs (G-CNN, D-CNN), without the comparison to (G-CNN, D-trans) and (G-trans and D-CNN). In this way, I am not sure which part the transformer-based structure is helpful.
> **A**:  Please find your requested results below, that we have had and just did not include in submission due to space limit. As we can see, changing into a transformer generator can already improve IS/FID, and matching that with another transformer discriminator boosts further. We will include this table into our final paper.
>
> |  Generator  | Discriminator |  IS  |  FID  |
> |:-----------:|:-------------:|:----:|:-----:|
> |   AutoGAN   |    AutoGAN    | 8.60 | 12.72 |
> | Transformer |    AutoGAN    | 8.82 |  9.89 |
> |   AutoGAN   |  Transformer  | 8.10 | 14.82 |
> | Transformer |  Transformer  | 9.02 |  9.26 |
>
>
> Reference:
> [a] Sharpness-aware minimization for efficiently improving generalization.
> [b] When Vision Transformers Outperform ResNets without Pretraining or Strong Data Augmentations.

---

> > ### Comment · Reviewer_M68S · 2021-09-06
> > **Post rebuttal update**
> >
> > I have carefully read the author's feedback. Many thanks for the author's efforts.
> > Most of my concerns are addressed. I will raise my rating from 5 to 6.

---

### Official Review · Reviewer_c687 · 2021-07-17

**Rating:** 7
**Confidence:** 4

**Summary:**

This work successfully takes an important step in bringing transformer architecture to the family of GAN models, so far dominated by convolutional architectures. Another contribution is the localised attention ('Grid Self-Attention') that addresses the problem of increased memory requirements of attention at higher-resolution feature map. The empirical evaluation shows competitive results on several benchmarks, including state-of-the-art scores on STL-10; however, it is missing an important benchmark - ImageNet.

Despite such limitation and missing references and comparisons to some relevant prior work (which hopefully could be addressed in rebuttal), this work seems to meet the quality and novelty requirements for NeurIPS. I am willing to improve the rating further if my concerns are addressed.

**Limitations And Societal Impact:**

The main limitation of this work is addressing only the unconditional generation problem, which has lower potential for applications. Conditional generation is important both inside and outside image domain and one of the major benchmark datasets, ImageNet, is not considered.

In broader impact, authors could acknowledge potential harmful uses of GANs, not only the positive ones.

**Main Review:**

The paper is generally clearly written, tackles a significant problem and proposes architectural novelties that will likely influence future works in generative modelling. Authors also made their code open, which will help reproducibility. I enjoyed reading a paper and think this work has potential to be among the most important ones submitted to NeurIPS this year; however, several issues should be addressed.

1. The Multi-Scale discriminator part is a bit obscure.
- 'multi-scale discriminator is designed to take varying size of patches' - how do they vary, what's the distribution of patch size?
- 'split the input images into three different sequences' - how is it split?
Sequence sizes and Figure 2 do not reflect either of these statements, my best guess is that the whole image was resized and input at each of the stages, but it's neither varying nor split. If I am correct, values of P are not reported, so it is very difficult to understand what really happens with the input.

2. In addition, Section 3.2 is missing relevant references: multi-scale discriminators have been in use for quite a while, both in image [4, 5, 6] and audio domain [7, 8], both with constant and random patch selection. Even though authors cite [5], they claim multi-scale discriminators to be their contribution 77: 'we also improve the discriminator with a multi-scale structure'. Number of stages and their composition is also not ablated.

3. The dataset choice seems somewhat peculiar given the claims that (i) the model design was focused on high resolutions and (ii) transformers are data-hungry, as compared to CNNs. Firstly, 'church' is neither most commonly used class out of LSUN categories, nor the one that contains most examples (relevance of 'smaller categories' notwithstanding, it's 126k vs 3M of popular 'bedrooms'). Secondly, the important ImageNet baseline is not addressed (which was studied both in unconditional [1] and conditional case [2, 3]). Authors explain the absence of conditional setup with simplicity argument, yet this is a significant limitation.

4. Overall, BigGAN [2] is almost not mentioned, even though it also applied self-attention.

5. Table 3. / Section 4.4
Authors discuss the issues with attention modules at higher-resolution maps, but do not provide FLOPs of various models, which would certainly help the reader have a clearer picture of those differences, and allow better comparison with CNN-based benchmarks. Parameter counts are also not provided.
The memory issues caused by some attention variants could potentially be solved e.g. by smaller batch sizes (or with micro-batching for fair comparison), or with ZeRO optimiser [9] (especially as all-experiments were carried in a multi-GPU setup).

Typos, wording, etc.:
- 63: 'more ambitious' - it's subjective and somewhat combative statement, suggest prior works were less ambitious. Maybe 'our goal required addressing additional new challenges'
- 203: non-overlapped->non-overlapping
- 248: describe->described, also e-8 -> e^{-8}
- 253: train->trains
- 320: fail-> fails

References:

[1] Mario Lucic, Michael Tschannen, Marvin Ritter, Xiaohua Zhai, Olivier Bachem, Sylvain Gelly, 'High-Fidelity Image Generation With Fewer Labels', ICML, 2019

[2] Andrew Brock, Jeff Donahue, Karen Simonyan, 'Large Scale GAN Training for High Fidelity Natural Image Synthesis', 2018

[3] Prafulla Dhariwal, Alex Nichol, 'Diffusion Models Beat GANs on Image Synthesis', 2021

[4]  Emily L Denton, Soumith Chintala, Arthur Szlam, and Rob Fergus. 'Deep generative image models using a Laplacian pyramid of adversarial networks' NeurIPS, 2015.

[5] Tero Karras, Timo Aila, Samuli Laine, Jaakko Lehtinen, 'Progressive growing of GANs for improved quality, stability, and variation' ICLR, 2018

[6] Han Zhang, Tao Xu, Hongsheng Li, Shaoting Zhang, Xiaogang Wang, Xiaolei Huang, Dimitris N Metaxas. Stackgan, 'Text to photo-realistic image synthesis with stacked generative adversarial networks' ICCV, 2017

[7] Mikołaj Bińkowski, Jeff Donahue, Sander Dieleman, Aidan Clark, Erich Elsen, Norman Casagrande, Luis C. Cobo, Karen Simonyan 'High Fidelity Speech Synthesis with Adversarial Networks', 2019

[8] Kundan Kumar, Rithesh Kumar, Thibault de Boissiere, Lucas Gestin, Wei Zhen Teoh, Jose Sotelo, Alexandre de Brebisson, Yoshua Bengio, Aaron Courville 'MelGAN: Generative Adversarial Networks for Conditional Waveform Synthesis', 2019

[9] Samyam Rajbhandari, Jeff Rasley, Olatunji Ruwase, Yuxiong He, 'ZeRO: Memory Optimizations Toward Training Trillion Parameter Models'


**Time Spent Reviewing:**

4

---

> ### Author Response · Authors · 2021-08-10
> **Response to Reviewer c687**
>
> **Q1**: The Multi-Scale discriminator part is a bit obscure. My best guess is that the whole image was resized and input at each of the stages, but it's neither varying nor split. If I am correct, values of P are not reported, so it is very difficult to understand what really happens with the input.
> **A**: The way of multi-scale splitting is very similar to the way of resizing and then splitting. However, the former one directly split the input with three different patch sizes. Take 256 x 256 input as an example, it is split with three different patch sizes 4, 8, and 16, resulting in three different sentences with their length equal to (64 x 64), (32x32), and (16x16), respectively. The value of P is reflected in Table 1-4 in our supplementary, where the input shape of the discriminator is equal to resolution dividing patch size. We agree the word “varying” is not precise as suggested by the reviewer, and we will reformulate this section in our main manuscript for better illustration.
>
>
>
> **Q2**: The number of stages is not ablated.
> **A**: During the rebuttal week, we conduct experiments on the LSUN-Church dataset to ablate the number of stages for our multi-scale discriminators. We start by adding the noise injection [22] technique at the beginning of the self-attention module, as we found it further improves the FID results on the LSUN-Church dataset from 8.94 to 7.48. After that, we evaluate three different groups with numbers of stages equal to 2, 3, and 4, respectively, while each stage contains three transformer blocks. As shown in the table below, fewer stages show inferior power to balance between capturing global contexts and local details, while more stages may result in overfitting issues. We will include more experimental results on other datasets in our main manuscripts.
>
> | Number of stage |  FID |
> |:---------------:|:----:|
> |        2        | 8.21 |
> |        3        | 7.48 |
> |        4        | 7.69 |
>
>
>
>
> **Q3**: Section 3.2 is missing relevant references, multi-scale has been proposed before.
> **A**: We thank the reviewer for the literature pointer. Although previous works already include many discussions on multi-scale discriminators, we are the first to introduce this technique to the vision transformer and demonstrate its effectiveness. We will make sure to carefully discuss this contribution in this clear context, and tone down its claim appropriately. Also, the contribution of this work is not just a single building block, but a completely new framework that pushes the limit of a purely transformer-based GAN and explores its potential power. We will cite the missing references in our main manuscript.
>
>
> **Q4**: Conditional image generation is missing.
> **A**: The current state-of-the-art conditional GAN employs many task-specific designs which are inapplicable for unconditional GAN, such as class-conditional normalization, projection discriminator, auxiliary classifier, and so on. This suggests that training a powerful conditional TransGAN may also require not only one, but a series of non-trivial adjustments. Therefore, we focus on pushing the limits of TransGAN in an unconditional setting. We agree that conducting evaluation on a conditional benchmark is also very important and will explore it further in our future work.
>
>
>
> **Q5**: The dataset choice seems somewhat peculiar given the claims that (i) the model design was focused on high resolutions and (ii) transformers are data-hungry, as compared to CNNs. Firstly, 'church' is neither the most commonly used class out of LSUN categories, nor the one that contains most examples (relevance of 'smaller categories' notwithstanding, it's 126k vs 3M of popular 'bedrooms').
> **A**: Our main testbed is conducted on the common benchmarks (Cifar-10, STL-10,CelebA, CelebA-HQ, and LSUN-Church), which cover different resolutions (from 32x32 to 256x256) and different environments (face and natural images). We firstly consider CelebA-HQ to push TransGAN to higher resolution tasks. However, the face image is homogeneous and less challenging compared to other natural images, which might not be enough to fairly evaluate the performance of TransGAN. Therefore, we consider another dataset LSUN to provide further evidence on evaluating our method. Although the LSUN-BED dataset contains much more examples (3M) to train TransGAN, it also requires more computational cost and training time to converge. Among the common categories of LSUN dataset (cat, church, bed, and car), “church” category provides a reasonable dataset size so that we are able to finish the experiments in an acceptable period. The specific training cost of each dataset is included in the supplementary. We are also preparing the training recipe for the LSUN-BED dataset. We will include it in our main manuscript later, and the corresponding codebase as well.
>
> **Q6**: BigGAN [2] and Imagenet are not considered.
> **A**:  As we illustrated in Q4, conducting conditional GAN evaluation requires many non-trivial adjustments for current TransGAN architecture, thus the conditional setting is not considered currently. Since the BigGAN method and ImageNet benchmark is often adopted in a conditional benchmark, we are not able to fairly compare them with our TransGAN architecture. Therefore, we choose the state-of-the-art unconditional GAN, StyleGAN-v2, as our main competitor. Although the reviewer mentioned that some previous works [1] conduct ImageNet evaluation in an unconditional setting, their main structure still requires another auxiliary classifier, and 10% labeled data is necessary to match SOTA performance. This may suggest that training an unconditional GAN on ImageNet dataset is still very challenging and needs more exploration before being adopted as a common benchmark. We agree that both the conditional setting and ImageNet evaluation are very important and we will explore it further in our future work.
>
>
> **Q7**: Table 3. / Section 4.4 Authors discuss the issues with attention modules at higher-resolution maps, but do not provide FLOPs of various models, which would certainly help the reader have a clearer picture of those differences, and allow better comparison with CNN-based benchmarks. Parameter counts are also not provided. The memory issues caused by some attention variants could potentially be solved e.g. by smaller batch sizes (or with micro-batching for a fair comparison), or with ZeRO optimizer [9] (especially as all-experiments were carried in a multi-GPU setup).
> **A**: We include the FLOPs cost and model size of each efficient variant of self-attention in the table below, where the resolution is set to be 256 x 256. The FLOPs cost of grid self-attention is less than the other two variants and the model size is smaller compared to the Nystrom self-attention. We thank the reviewer for suggesting ZeRO optimizer and smaller batch size as alternative solutions and will try them in our future work.
>
> |         Methods        | FLOPs/G | Params/M |
> |:----------------------:|:-------:|:--------:|
> |   Axis Self-attention  |  62.18  |  119.18  |
> | Nystrom Self-Attention |  82.98  |  150.29  |
> |   Grid Self-attention  |  48.04 |  119.18  |
>
>
> **Q8**: Some typos need correction. On line 63, 'more ambitious' - it's a subjective and somewhat combative statement, suggests prior works were less ambitious. Maybe 'our goal required addressing additional new challenges'
> **A**: We appreciate the reviewer for providing these constructive suggestions. We agree that ‘Ambitious’ is too subjective and “new challenges” is a more precise illustration here. We will correct them in our main manuscript.
>
> **Q9**: Authors could acknowledge potential harmful uses of GANs, not only the positive ones.
> **A**: We will include more potential harmful uses of GANs in our main manuscripts, including the spreading of fake news, copyright infringement, and privacy concerns.

---

> > ### Comment · Reviewer_c687 · 2021-08-11
> > **Thank you for clarifications.**
> >
> > I would like to thank authors for clarifications; I am looking forward to see the updated version of the paper.
> >
> > As for Q5, I would like to clarify that I am not exactly questioning the sufficiency of chosen datasets (5 different ones seem to be very reasonable), but the inconsistency between them and the mentioned statements. Although I agree that there are challenges with training on 3M examples of 256x256 images (LSUN bedrooms), if the model design was 'focused on high resolutions' and used 'data-hungry architecture', one should be prepared to face those obstacles. It seems that the current setup for LSUN-church would allow 4-5 full epochs on LSUN-bedrooms, even if that's less than in Style-GAN benchmark (70M examples; >20 epochs), reporting results with such a setup would be very much appreciated. In any case, I am very keen to hear the authors are preparing to train their model on this dataset.

---

### Official Review · Reviewer_QfD2 · 2021-07-19

**Rating:** 6
**Confidence:** 5

**Summary:**

This paper has presented TransGAN: a purely Transformer-based architecture for Generator and Discriminator in GANs. To solve the huge computation complexity of self-attention in high-resolution, this paper introduces grid-self-attention. Also, to solve the unstable training process of GANs, this paper introduces several techniques such as multi-scale Discriminator, and modified normalization. The experiments are conducted on various datasets.

**Limitations And Societal Impact:**

The authors have addressed the limitations and potential negative societal impact.

**Main Review:**

Strength:
1. This paper tries to build a pure Transformer-based architecture for the Generator and Discriminator in GANs. previous works have demonstrated the effectiveness of Transformer-based architecture in classification, object detection, and segmentation tasks. So the idea of Transformer-based models in GANs is well motivated and intuitive.

2. Building Transformer-based architectures for Generator and Discriminator in GANs may have many challenges, like unstable training process, the huge computation complexity of self-attention in high resolutions. This paper proposes several methods to solve these problems.

Weakness:
1. The experimental results of this paper are weak and some comparisons are unfair obviously. For example, the comparison in Table 1 is unfair since the FID of StyleGAN v2 and StyleGAN v2+DiffAug on CIFAR-10 are calculated in 10k samples in the original paper[68]. However, in this paper, the FID is calculated with 50k samples. Applying different numbers of samples for calculating FID can yield different FID scores, more samples can yield better performance. Also, the original result of StyleGAN-v2 on LSUN-Church(256$\times$256) is $\textbf{3.86}$ for FID score, while TransGAN achieves 8.94, indicating the current results of TransGAN are significantly worse than StyleGAN2.

2. The strength of applying the self-attention module for Generator and Discriminator in GANs is the capability of capturing long-range dependencies, However, the grid self-attention module is also a local operator, which may show limitations in capturing long-range dependencies. So I am wondering whether the idea of applying the grid self-attention module may restrict the capability of the Generator?

**Time Spent Reviewing:**

6 hours

---

> ### Author Response · Authors · 2021-08-10
> **Response to Reviewer QfD2**
>
> **Q1**: The comparison in Table 1 is unfair since the FID of StyleGAN-v2 and StyleGAN-v2+DiffAug on CIFAR-10 are calculated in 10k samples in the original paper [68]. However, in this paper, the FID is calculated with 50k samples.
> **A**: We follow the suggestion to evaluate our method on 10k samples but do not observe obvious performance degradation. The best FID reaches 9.34 which still outperforms StyleGAN-v2 with DiffAug. Moreover, we also tried the "noise injection" trick [22, 40] at the beginning of the self-attention module in our generator, and observe that the best FID can be further improved from 9.34 to 8.96. As a result, it outperforms StyleGAN-v2 with a large margin on the CIFAR-10 dataset.
>
> **Q2**:  The original result of StyleGAN-v2 on LSUN-Church(256256) is 3.86 for FID score, while TransGAN achieves 8.94.
> **A**: StyleGAN-v2 employs many heavily engineered "tricks" such as noise injection, equalized learning rates, weight demodulation, lazy regularization, and mini-batch standard deviation, to stabilize the training process and improve image quality. While similar training techniques on ConvNet-based GANs have been explored for several years, we believe that TransGAN will be an encouraging starting point for transformer-based GANs and there is still a large room to explore further. For instance, during the rebuttal week, we apply the noise injection technique [22, 40] at the beginning of the self-attention module in our transformer generator and find it further improves the FID result on LSUN-Church (256 x 256) from 8.94 to 7.48. We will add more ablation studies on those training tricks in our main manuscript.
>
>
> **Q3**: The strength of applying the self-attention module for Generator and Discriminator in GANs is the capability of capturing long-range dependencies, However, the grid self-attention module is also a local operator, which may show limitations in capturing long-range dependencies. So I am wondering whether the idea of applying the grid self-attention module may restrict the capability of the Generator?
>
> **A**: The Grid Self-attention is only applied in the stages where the resolution is larger than 32 x 32. Thus the proposed architecture can capture both the long- and short-range dependencies. Since it is always a trade-off between efficiency and performance, we also compare the proposed Grid Self-attention with two popular efficient variants (Nystrom and Axis Self-attention), to prove its effectiveness. As shown in Table 3, the grid self-attention can largely outperform other competitors.

---

### Author Response · Authors · 2021-08-10
**Summary of Author's Response**

We thank all reviewers for their insightful and constructive suggestions. We are glad that reviewers found (1)  Building a GAN model using only transformer blocks is well motivated and intuitive. (Reviewer QfD2); (2) Ablation experiments on each architecture or training algorithm component are very thorough and convincing (Reviewer PbKe); (3) Grid self-attention makes it possible to apply transformers to large-scale GAN training (Reviewer M68S ); (4) The paper is easy to follow and has the potential to be among the most important ones submitted to NeurIPS this year. (Reviewer c687)


We have addressed all the questions that the reviewers posed with additional experimental results. We will carefully modify our main manuscript later, following those suggestions.

---

### Decision · Program_Chairs · 2021-09-27

**Decision:**

Accept (Poster)

**Comment:**

This submission demonstrates that GANs with transformer-based architectures can learn to generate high-quality and high-resolution images, achieving strong results competitive with (or in some cases better than) convolution-based GANs on a variety of benchmark datasets.

Reviewers were generally in agreement that the paper is well-executed and does a nice job of tackling the scale-related challenges that arise from applying transformers to high resolution images. Although not all results are state-of-the-art, the paper makes use of very non-traditional architectures and as such doesn’t benefit as much from the tricks of the trade that have been developed and refined over several years of progress in convolutional GAN models, so this should not preclude acceptance. Another criticism that came up in review was the lack of results in the class-conditional setting (e.g. ImageNet). I would agree with the authors that it’s fair to save this for future work especially given the thoroughness of the evaluation in unconditional settings. Reviewers pointed out certain aspects of the method that weren’t clear and/or well-ablated in the original submission (e.g. what happens if transformers are used for only G and/or only D?) and the authors provided thorough responses including additional results/ablations. These results and clarifications should of course be included in the camera-ready version of the paper.

Given its strong image generation results using non-traditional architectures of current interest to much of the NeurIPS audience (with findings/ideas potentially useful beyond image generation), I recommend accepting the submission.